Investigating the epidemiological relevance of secretory otitis media and neighboring organ diseases through an Internet search

Guo Cheng 1
Pan Linlin 1
Chen Ling 1
Xie Jinghua 1
Liang Zhuozheng 2
Huang Yongjin 1
He Long eyhelong@scut.edu.cn 1
1 Department of Otorhinolaryngology Head and Neck Surgery, Guangzhou First People’s Hospital, the Second Affiliated Hospital of South China University of Technology , Guangzhou , Guangdong Province , China
2 Intensive Care Unit, The First People’s Hospital of Foshan , Foshan , Guangdong Province , China
Kabir Russell
Electronic publication date: 2024 Mar 5
Publication date: 2024
Volume: 12
Electronic Location ID: e16981
Received 2023 Sep 20; Accepted 2024 Jan 29
Copyright: ©2024 Guo et al.
Copyright year: 2024
Copyright holder: Guo et al.
License: This is an open access article distributed under the terms of the Creative Commons Attribution License, which permits unrestricted use, distribution, reproduction and adaptation in any medium and for any purpose provided that it is properly attributed. For attribution, the original author(s), title, publication source (PeerJ) and either DOI or URL of the article must be cited.
License URL: https://creativecommons.org/licenses/by/4.0/

Keywords: Secretory otitis media, Baidu Index, COVID-19, Internet, Big data

Funding: The authors received no funding for this work.

==============================
Background

This study examined the epidemiological correlations between secretory otitis media (SOM) and diseases of neighboring organs. We measured changes in disease incidences during the 2020 COVID-19 pandemic using Internet big data spanning from 2011 to 2021.

Methods

This study used the Baidu Index (BI) to determine the search volume for the terms “secretory otitis media (SOM)”, “tonsillitis”, “pharyngolaryngitis”, “adenoid hypertrophy (AH)”, “nasopharyngeal carcinoma (NPC)”, “nasal septum deviation (NSD)”, “rhinosinusitis”, “allergic rhinitis (AR)”, and “gastroesophageal reflux disease (GERD)” in Mandarin from January 2011 to December 2021. The correlations between these terms were analyzed using Spearman’s correlation coefficients. The results were compared search data from 2019 and 2021 to assess the effects of isolation on SOM in 2020.

Results

The seasonal variations trends of SOM and other diseases coincided well (P < 0.05), except for AR. During the 11-year timeframe, the monthly searches for rhinosinusitis, NSD, tonsillitis, pharyngolaryngitis, and NPC were statistically correlated with SOM (R = 0.825, 0.594, 0.650, 0.636, 0.664, respectively; P < 0.05). No correlation was found between SOM and AR, SOM and AH, or SOM and GERD (R =  − 0.028, R = 0.259, R = 0.014, respectively, P > 0.05). The total search volumes for SOM, rhinosinusitis, NPC, and AH decreased in 2020 compared to 2019.

Discussion

SOM exhibited a discernible epidemiological connection with rhinosinusitis, nasal septal deviation (NSD), tonsillitis, pharyngolaryngitis, and nasopharyngeal carcinoma (NPC). A decrease in public gatherings was observed to effectively reduce the incidences of SOM. This underscores the pivotal role of social measures in influencing the prevalence of SOM and emphasizes the intricate interplay between SOM and various associated health factors, with implications for public health strategies.

Introduction

Secretory otitis media (SOM) is a frequently diagnosed disease characterized by middle-ear effusions, which reduce tympanic membrane mobility, and increase hearing loss and ear swelling (Chen et al., 2023). Children have a higher incidence of SOM than adults, with about 90% of preschoolers experiencing at least one episode. Most children with SOM will self-heal; however, chronic SOM may lead to speech retardation, a decline in academic performance, and other problems. The pathogenesis of SOM is related to adenoid hypertrophy (AH), nasopharyngeal carcinoma (NPC), rhinosinusitis, allergic rhinitis (AR) (Rosenfeld et al., 2016), nasal septum deviation (NSD), gastroesophageal reflux disease (GERD), and so on (Sone et al., 2013).

Epidemiological data are typically provided by clinicians and preventive health professionals to health authorities who collate and disclose the information to the public. This method of sharing information consumes a large amount of time and resources, leading to the delayed surveillance of emerging infectious diseases. However, the growth of the Internet has allowed people to easily and quickly access health information, and provides a new way to detect and monitor disease outbreaks (Eysenbach, 2009). Internet search data can provide real-time information and has good consistency with traditional healthcare surveillance systems. Internet search data has been used to investigate the epidemic characteristics of diseases in recent years (Milinovich et al., 2014). For example, using Internet search data, scholars found that influenza, depression, sleep-disordered breathing, and other diseases have a seasonal onset (Ingram, Matthews & Plante, 2015), and AH and rhinosinusitis are correlated (Yang et al., 2022). Moreover, Internet search data can also be applied to predict the incidence of HIV and syphilis (Huang et al., 2020).

Baidu and Google are currently the world’s largest search engines. Baidu is the most dominant search engine since Google pulled out of China in 2010 and researchers who have studied the use of Internet data have verified that the Baidu Index (BI) is reliable as the incidence index of diseases (Yang et al., 2022).

The novel coronavirus disease 2019 (COVID-19), a highly contagious respiratory disease, emerged in Wuhan, Hubei Province, China in December 2019. The Chinese government locked down Wuhan from January to April 2020 to prevent the spread of COVID-19 and urged people to stay at home. This real-life scenario provides an opportunity to investigate the effect of isolation on the incidence of diseases such as SOM, rhinosinusitis, NSD, AR, AH, NPC, tonsillitis, pharyngolaryngitis, and GERD.

The study employed a relatively novel methodology-the Internet search-to enhance our understanding of the epidemiology of SOM. This approach provides an alternative perspective for investigating the relationship between SOM and adjacent organs. The influence of isolation measures on the occurrence of SOM and diseases in neighboring organs was also investigated amid the COVID-19 pandemic in 2020, spanning from 2011 to 2021.

Materials & Methods

Data from the Baidu Index

Baidu is used by a majority of Chinese Internet searchers for a wide range of search queries. Therefore, the Baidu Index may reflect the behavioral characteristics of the Chinese populace. We conducted searches on the BI using Chinese characters for the terms “secretory otitis media”, “tonsillitis”, “pharyngolaryngitis”, “adenoid hypertrophy”, “nasopharyngeal carcinoma”, “nasal septum deviation”, “rhinosinusitis”, “allergic rhinitis” and “gastroesophageal reflux disease” from January 2011 to December 2021. Simultaneously, to observe the influence of COVID-19 isolation measures, we incorporated the initial 5 months of the aforementioned data for the years 2019, 2020, and 2021 within the study’s timeframe, facilitating a comparison of search volumes during the corresponding period across these years. The variables and calculations for our search were calculated as follows (Table S1): (1) The average daily search volume of every term was original from BI. (2) The monthly search volume was calculated as the average daily search volume for the month multiplied by the number of days in that month. (3) The annual search volume was obtained by summing up the monthly search volumes for the entire year. (4) The percentages of the monthly search volume were calculated by dividing the monthly search volume for the year by the annual search volume. (5) The percentages of average monthly search volume were obtained by summing the monthly search volumes for the same month throughout the year and dividing the result by the total annual search volume.

Data analysis

The main variables were the percentages of monthly search volume and the average monthly searches for tonsillitis, pharyngolaryngitis, AH, NPC, NSD, rhinosinusitis, AR and GERD. Statistical analyses were performed using SPSS Statistics version 26.0 (IBM Corp., Armonk, NY, USA). Based on the results of the Shapiro–Wilk normality test, not all continuous variables in this study followed a normal distribution. Thus, Spearman’s correlation coefficients were used to analyze the correlation between variables. Paired comparisons among the first 5 months of the data in 2019, 2020, and 2021 were performed with nonparametric Friedman test. P <0.05 was considered significant. Microsoft Excel Software and RStudio 2022 (RStudio, Inc., 2022) were used to draw line graphs and heatmaps.

Results

A total of 132 monthly search volume values were obtained. In this study, tonsillitis, acute tonsillitis, and chronic tonsillitis were merged into “tonsillitis”; likewise, “acute pharyngitis”, “chronic pharyngitis”, “pharyngitis”, “laryngitis”, “chronic pharyngolaryngitis”, and “pharyngolaryngitis” were included in “pharyngolaryngitis”. SOM was compared with the other eight search terms to determine the relationship between SOM and nasal disease, laryngeal disease, oropharyngeal diseases, and GERD. The correlation of the nine search terms from 2011 to 2021 are presented in Fig. 1. Except for AR, the trendlines of SOM and the other seven diseases coincided well. As shown in Fig. 1, the seasonal variations of the trendlines between SOM and NSD, tonsillitis, and pharyngolaryngitis were strongly correlated (R = 0.584, 0.588, 0.502, respectively, all P <0.05), while the relationship between SOM and rhinosinusitis, NPC, AH, GERD was weak (R = 0.475, 0.358, 0.351, 0.339, respectively, all P <0.05).

Figure 1 Monthly search volume (in percentage).

(A) SOM and nasal diseases (AR, rhinosinusitis and NSD). (B) SOM and nasopharyngeal diseases (NPC and AH). (C) SOM and laryngeal diseases (tonsillitis and pharyngolaryngitis). (D) SOM and GERD during 2011–2021. Except for AR, the trendlines of SOM and the other seven diseases were highly coincidental. AR, allergic rhinitis; NSD, nasal septum deviation; SOM, secretory otitis media; NPC, nasopharyngeal carcinoma; AH, adenoid hypertrophy; GERD, gastroesophageal reflux disease.

The average monthly search volumes for the nine keywords during the 11-year timeframe are shown in Fig. 2 and Table S2. The monthly searches for rhinosinusitis, NSD, tonsillitis, pharyngolaryngitis, and NPC showed a similar trend, which correlated significantly with SOM (R = 0.825, 0.594, 0.650, 0.636, 0.664, respectively, all P <0.05). As shown, the highest and lowest average monthly search volumes for SOM, rhinosinusitis, NSD, tonsillitis, and pharyngolaryngitis were in February and March. Furthermore, the curve was more gradual from April to December. However, no correlation was detected between SOM and AR, SOM and AH, or SOM and GERD (R = −0.028, 0.259, 0.014, respectively, all P >0.05).

Figure 2 Average monthly search volume.

(A) SOM and nasal diseases (AR, rhinosinusitis and NSD). (B) SOM and nasopharyngeal diseases (NPC and AH). (C) SOM and laryngeal diseases (tonsillitis and pharyngolaryngitis). (D) SOM and GERD during 2011–2021. SOM and rhinosinusitis, NSD, tonsillitis, pharyngolaryngitis, NPC all drastically decline from January to February and slowly decline from March to September and then increase from February to May and from November to December. AR, allergic rhinitis; NSD, nasal septum deviation; SOM, secretory otitis media; NPC, nasopharyngeal carcinoma; AH, adenoid hypertrophy; GERD, gastroesophageal reflux disease.

As shown in Fig. 3 and Table S3, the search volumes for SOM, rhinosinusitis, NPC, and AH decreased during the first 5 months of 2020 compared with the same period in 2019 (P <0.05). In addition, the total search volumes for NSD and pharyngolaryngitis were found to decrease during the first 5 months of 2021 compared with 2019 (P <0.05). For AR, tonsillitis, and GERD, no statistical correlation was found between SOM and any one of them (P >0.05). The correlation between SOM and other search volumes can be observed intuitively in the heatmaps in Figs. 4 and 5.

Figure 3 Comparison of the search volume during the first five months of 2019, 2020 and 2021.

(A) SOM. (B) AR. (C) Rhinosinusitis. (D) NSD. (E) NPC. (F) AH. (G) Tonsillitis. (H) Pharyngolaryngitis. (I) GERD. The total search volumes for SOM, rhinosinusitis, NPC and AH, pharyngolaryngitis decreased during the first five months of 2020 compared with those months in 2019 (P > 0.05). The total search volumes for NSD and pharyngolaryngitis were found to decrease during the first 5 months of 2021 compared with 2019 (P < 0.05). AR, allergic rhinitis; NSD, nasal septum deviation; SOM, secretory otitis media; NPC, nasopharyngeal carcinoma; AH, adenoid hypertrophy; GERD, gastroesophageal reflux disease.

Figure 4 The heatmap shows the relationship between SOM and other eight diseases in monthly volume.

The search variations for SOM and rhinosinusitis, NSD, AH, NPC, tonsillitis, pharyngolaryngitis, GERD are significantly relevant (R = 0.475, 0.584, 0.351, 0.358, 0.358, 0.588, 0.502, 0.339, respectively, all P < 0.05). The search variations for SOM and AR are not related (R = 0.082, P > 0.05). AR, allergic rhinitis; NSD, nasal septum deviation; SOM, secretory otitis media; NPC, nasopharyngeal carcinoma; AH, adenoid hypertrophy; GERD, gastroesophageal reflux disease.

Figure 5 The heatmap shows the relationship between SOM and other eight diseases in average month volume.

The search variations for SOM and rhinosinusitis, NSD, tonsillitis, pharyngolaryngitis, NPC are strongly relevant R = 0.825, 0.594, 0.650, 0.636, 0.664, P < 0.01, respectively. No statistical relationship was found between SOM and AR, AH or GERD (R = –0.028, 0.259, 0.014, respectively, all P > 0.05). AR, allergic rhinitis; NSD, nasal septum deviation; SOM, secretory otitis media; NPC, nasopharyngeal carcinoma; AH, adenoid hypertrophy; GERD, gastroesophageal reflux disease.

Discussion

This study explored the correlation and seasonal prevalence of SOM with rhinosinusitis, AH, NPC, AR, NSD, pharyngolaryngitis, and GERD by using the BI. The correlation was further verified by showing that isolation during the COVID-19 pandemic reduced the incidence of rhinosinusitis and AH. Our results indicated that the monthly search volume of SOM was statistically consistent with the monthly search volume of rhinosinusitis, AH, NPC, NSD, pharyngolaryngitis, tonsillitis, and GERD. The monthly searches for SOM decreased in parallel with those for rhinosinusitis, AH, and NPC during the COVID-19 isolation period from January to May 2020. However, there was no correlation between SOM and GERD, AR, NSD, pharyngolaryngitis, and tonsillitis.

With the popularization of the Internet, an increasing number of people can access medical and health knowledge through an Internet search. Therefore, data from online surveillance tools may be used to predict disease prevalence and analyze epidemic characteristics. Compared to traditional healthcare surveillance systems, surveillance systems based on Internet search data are more convenient, economical, and fast. Mounting evidence has shown that Internet search data can be used to monitor and predict influenza, AIDS, gonorrhea, syphilis, and other infectious diseases (Huang et al., 2020). In addition, the sensitivity and specificity of monitoring or forecasting results based on Internet search data are comparable to those of traditional surveillance systems.

The correlation between SOM and rhinosinusitis is well-established (Rosenfeld et al., 2016). Finkelstein et al. (1994) showed that 66% of adults with SOM had rhinosinusitis, especially ethmoid sinus lesions. Previous studies reported that 23% of patients with chronic rhinosinusitis suffered from SOM (Dang & Gubbels, 2013). Some research exploring the correlation between SOM and nasal polyps indicated that the incidence of SOM was high in severe nasal polyps, such as patients with asthma and aspirin intolerance (Taylor, Evans & Hope, 1974). However, the specific pathogenesis is still unclear. The scholar analyzed two possibilities: mechanical blockage of the eustachian tube and simultaneous inflammation of the mucous membrane of the middle ear and nasal passages (Parietti-Winkler et al., 2009). Surprisingly, this study also discovered that the incidence of rhinosinusitis coincided with SOM. A multicenter prospective study confirmed that patients with sinusitis with eustachian tube dysfunction (ETD) typically experience improved eustachian tube function after surgery (Chen et al., 2021). However, Daval et al. (2018) suggested no significant correlation between the incidence of SOM and the severity of nasal polyps and nasal blockage. SOM may develop or persist even after nasal congestion improves with surgical or conservative medical treatment. Therefore, it was speculated that the correlation between sinusitis and SOM was mainly due to hypersecretion of the nasal mucosa and middle ear mucosa stimulated by inflammation.

Adenoids are immune organs in childhood that gradually increase until 7–10 years old, but then begin to shrink with age. However, repeated inflammatory stimulation can lead to AH. The correlation between the incidence of SOM and AH has been verified (Low et al., 1997). Our study discovered that the occurrence of SOM and AH were consistent, and the difference was statistically significant. Many studies have shown that the nasopharyngeal bacteria of patients with SOM have homology with the middle ear effusion bacteria. Obstruction of the eustachian tube (ET) caused by adenoid enlargement led to the accumulation of nasopharyngeal secretions and microorganisms. Retrograde microbial infection is the main cause of SOM caused by AH (Tawfik et al., 2016; Pagella et al., 2010). Pharyngolaryngitis and tonsillitis are often accompanied by bacterial infection in the pharynx. Research has shown that SOM is associated with pharyngolaryngitis, and is not only related to retrogressive infection of pharyngeal bacteria along ET but may also be related to mucosal swelling around ET and ETD caused by repeated inflammatory stimulation. However, the correlation between adenoid enlargement and mechanical obstruction of ET and SOM remains ambiguous. Relevant studies have demonstrated that the mucous membrane of the middle ear has the same secretory function as the nasopharynx (Marple, 2010). When inflammation stimulates adenoid enlargement, it may also induce hypersecretion of middle ear mucosa, subsequently leading to SOM (Hsin et al., 2013; Young, 2019).

NPC is a common malignant tumor occurring in the nasopharynx, usually in the pharyngeal recess (Chen et al., 2019). Due to tumor compression and invasion, the function of ET is affected, and patients with NPC are prone to have SOM. However, studies have shown that for many newly diagnosed NPC, the etiology of ETD is still unknown (Su, Hsu & Chee, 1993). Mo et al. (2016) evaluated the ET function of NPC patients by Valsalva maneuver combined with MRI and found that ETD was significantly more common when tumors involved structures near the ET, especially the tensor veli palatine muscle. It has also been suggested that SOM is caused by an injury to the nerve that innervates the tensor veli palatine muscle. The primary treatment for NPC is radiotherapy (Wei & Sham, 2005). Radiotherapy may cause many local complications, such as otitis media, sensorineural hearing loss, and rhinosinusitis. Although modern intensity-modulated radiotherapy can reduce the incidence of these complications, researchers have shown that SOM is still a common late toxicity of the middle ear after radiotherapy (Hsin et al., 2013). Among the susceptibility factors to post-irradiation SOM, rhinosinusitis, and ETD have been fully confirmed. The nasopharyngeal and middle ear bacteriology of patients with post-irradiation sinusitis and SOM have homology, and SOM is alleviated by nasal irrigation. This indicates that there may be a correlation between SOM and bacterial infection from the paranasal sinuses after irradiation (Hsin et al., 2016b). Rinsing the nasal cavity with normal saline can relieve the symptoms of rhinosinusitis and SOM (Liang et al., 2008). However, Hsin et al. (2016a) found that post-irradiation SOM is not related to rhinosinusitis. They also believed that nasal irrigation had no effect on alleviating SOM symptoms, and incorrect nasal irrigation will cause retrograde movement of nasal microorganisms to the middle ear and aggravated SOM (Hsin et al., 2016a).

NSD may be associated with middle ear diseases. Ural, Minovi & Çobanoğlu (2014) found that 89.4% of patients with NSD had middle ear disease, and 50.4% of them had otitis media on the same side as the nasal congestion. Our study found a correlation between the incidence of NSD and SOM. Sub-mucous resection (SMR) of the nasal septum can improve the negative pressure regulation of the middle ear in patients with septal deviation and ETD (Upadhya & Datar, 2014). The relationship between septal deviation and SOM is ET. However, it is not clear how septal deviation affects the function of ET. Most studies suggest that NSD affects the function of ET by affecting the airflow in the nasopharynx (Maier & Krebs, 1998). When air flows through the narrow space, the velocity becomes faster and the laminar flow becomes turbulent, resulting in high local negative pressure and impaired ET function. Patients with nasal congestion may also develop greater negative pressure during strenuous breathing. Patients with NSD and ETD had disordered nasal airflow before SMR, and the middle ear cavity could not be inflated. After SMR, the nasopharyngeal airflow disorder disappeared, and the middle ear cavity could be well-inflated (Upadhya & Datar, 2014). Low & Willatt (1993) found that the difference in bilateral nasal patency caused by NSD was negatively correlated with the negative pressure in the middle ear on the side of the blocked nasal passage. Correcting this asymmetric nasal patency through SMR could improve the corresponding negative pressure in the middle ear. Therefore, they hypothesized that the asymmetry of bilateral nasal airflow caused by the NSD leads to turbulence in the nasopharynx, making it easier for microorganisms to deposit in the nasopharynx and around the pharyngeal orifice of ET, resulting in inflammation and mechanical obstruction around the ET. In addition, turbulence dries the air and affects the function of the mucous layer in the nasopharynx and ET. However, other studies have shown that NSD has little impact on the function of ET, and only severe deviation can affect the middle ear function.

Gastroesophageal reflux (GER) is the reflux of stomach contents into the esophagus. GERD is characterized by the abnormal reflux of stomach contents into the esophagus causing symptoms such as heartburn or complications such as esophageal injury (Dent et al., 1999). GERD not only causes esophageal lesions but is also associated with many diseases of the ear, nose, and throat (Ozmen et al., 2008; Lechien et al., 2021). Although GERD has been listed as one of the pathogenesis factors of SOM, the pathophysiological mechanism remains unclear. A Korean study showed that adults with GERD had a 1.84 times higher prevalence of chronic SOM than those without GERD (Yeo, Kim & Lee, 2021). Our study found that the incidence of SOM and GERD had the same trend. Previous studies found that many middle ear effusions of SOM consisted of pepsin/pepsinogen and hydrochloric acid (HCl), with gastric juice as the main component of the effusions. It was speculated that the inflammatory stimulation of pepsin/pepsinogen and HCl on the middle ear mucosa led to mucosal edema and hypersecretion. The damage to the ET was the key factor for SOM. It has been verified in rat experiments that pepsin and HCl can affect the function of ET and mucociliary clearance (White et al., 2002). Pepsinogen can only be activated into bioactive pepsin under an acidic environment, but the pH of the middle ear effusion is alkaline, and the pepsinogen in the middle ear effusion cannot be activated. However, no relevant study has confirmed the causal relationship between pepsin/pepsinogen in middle ear effusions and SOM.

AR refers to symptoms such as nasal congestion, itching, sneezing, and runny nose caused by allergies to certain substances. The association between AR and SOM is controversial. It has been established that AR is a risk factor for the pathogenesis of SOM. Some researchers have found that allergen exposure can induce ET obstruction and SOM symptoms in AR patients (Skoner, Doyle & Fireman, 1987). In addition, the finding of effector T helper 2 cells, whose expression is significantly higher in allergic patients, in the middle ear effusion of SOM patients indicates that allergic inflammation may be also present in the middle ear of SOM (Nguyen et al., 2004). In response to these findings, some studies treated patients with AR and SOM with nasal steroid sprays and azelastine hydrochloride and found it could relieve patients’ SOM symptoms (Bhargava & Chakravarti, 2014). The mucosa of the nasal cavity and the middle ear are contiguous, and both can produce an allergic inflammatory response. However, in the study of nasal polyps and SOM, some researchers found that the risk of SOM in patients with specific nasal polyps was not different from that in patients with non-specific nasal polyps, denying the role of allergic reactions in SOM (Parietti-Winkler et al., 2009). In this study, the incidence trend of SOM was not consistent with that of AR. Overall, the correlation between allergic diseases and SOM needs to be further studied.

Nasal symptoms in COVID-19 are similar to those of AR and sinusitis, increasing the complexity of the diagnosis. However, COVID-19 did not aggravate sinonasal symptoms in patients with sinusitis and AR (Ferreli et al., 2020; Marin et al., 2022). Yang et al. (2022) found that the incidence of AH and sinusitis decreased during the COVID-19 epidemic, which may be related to reduced gatherings and mask use during the outbreak (Huang et al., 2020). Marom et al. (2021) noted that the incidence of SOM decreased during the COVID-19 pandemic. This effect may be linked to lockdowns, quarantines, and fewer non-emergency visits due to fear of contracting the virus in outpatient clinics. In this study, the incidence of NPC, AH, and rhinosinusitis all coincided with a decline in the incidence of SOM during this period. The reduction of crowd gatherings and mask use greatly reduced the spread of microorganisms during COVID-19. However, there is weak evidence to support the impact of isolation due to COVID-19 on SOM and neighboring organ diseases in the first five months of 2021 when considering the repeated local epidemics of COVID-19 in China. SOM and lesions of the surrounding structures also decreased during this period, and it is well-known that microorganisms may play a crucial role in the pathogenesis of secretory otitis media and peripheral organ lesions.

Notwithstanding, the current study has certain limitations. Firstly, since rhinosinusitis and pharyngolaryngitis include acute, chronic, and other types, it is necessary to account for these diverse subtypes. However, it may be difficult to distinguish between the subtypes of different illnesses when searching for symptoms, thus, the keywords “tonsillitis”, “acute tonsillitis” and “chronic tonsillitis” were combined into “tonsillitis”, and “acute pharyngitis”, “chronic pharyngitis”, “pharyngitis”, “chronic pharyngolaryngitis”, and “pharyngolaryngitis” were combined into “pharyngolaryngitis”, making the study of subtypes complicated. Secondly, Internet search engines users are typically younger people. Our study may have an age bias, as children, the elderly, and those in underdeveloped areas rarely use search engines. Thirdly, the fluctuations in BI search behaviors may be influenced by media reports and social propaganda events. Finally, the symptoms of these diseases have may overlap, and the precise motivations behind these search behaviors are not able to be determined. Further big data analysis or more accurate data is needed to improve the reliability of utilizing the search query data as the monitoring method. With the further popularization of the Internet and the spread of medical science, insufficient clinical data may be effectively offset by Internet-based surveillance systems, thus providing a novel tool for monitoring disease morbidity and exploring the relationships among diseases.

Conclusions

During seasonal variations SOM was found to be strongly correlated with NSD, tonsillitis, and pharyngolaryngitis, and weakly correlated with rhinosinusitis, AH, NPC, and GERD. It was not significantly correlated with AR. A decrease in public gatherings significantly reduced the morbidities of SOM, rhinosinusitis, NPC, and AH.

Supplemental Information

Data S1 Raw data

Table S1 Variables and calculations

The variable from Baidu index: average daily search volume (D) of every term. 2) Monthly search volume (M) is calculated as the average daily search volume for the month multiplied by the number of days in that month (n). (e.g., The average daily search volume of February 2011 is 576 times, then the monthly search volume in February 2011 equals 16128 times (576*28 = 16128). 3) Annual search volume (Y) is then obtained by summing up the monthly search volumes for the entire year. Results are presented as percentages: 4) The percentages of monthly search volume (P) are calculated by dividing the monthly search volume for the year by the annual search volume. 5) The percentages of average monthly search volume (Pm) are obtained by summing the monthly search volumes for the same month throughout the year and dividing the result by the total annual search volume.

Table S2 Average monthly search volume

Table S3 The search volume during the first 5 months from 2019 to 2021

Additional Information and Declarations

Competing Interests

Author Contributions

Data Availability

The authors declare there are no competing interests.

Cheng Guo conceived and designed the experiments, prepared figures and/or tables, and approved the final draft.

Linlin Pan performed the experiments, analyzed the data, prepared figures and/or tables, and approved the final draft.

Ling Chen performed the experiments, prepared figures and/or tables, authored or reviewed drafts of the article, and approved the final draft.

Jinghua Xie conceived and designed the experiments, authored or reviewed drafts of the article, and approved the final draft.

Zhuozheng Liang conceived and designed the experiments, authored or reviewed drafts of the article, and approved the final draft.

Yongjin Huang analyzed the data, authored or reviewed drafts of the article, and approved the final draft.

Long He conceived and designed the experiments, performed the experiments, authored or reviewed drafts of the article, and approved the final draft.

The following information was supplied regarding data availability:

https://index.baidu.com

Raw data used uploaded in Supplemental Files.

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
