# Peer review of "Investigating the epidemiological relevance of secretory otitis media and neighboring organ diseases through an Internet search"

_PeerJ, doi:10.7717/peerj.16981_

## Round 0.1 · original submission · Major Revisions

Thanks for considering the Peerj journal. Please address the queries raised by the reviewers. Additionally, address the following queries-

1. No keywords are mentioned in the abstract.

2. Could you tell me the reason for doing this research?

3. What will be the impact of the findings?

4. Why the search timeline was only for the last ten years?

5. Line 84-86, not clear, needed further clarification. If possible, could you show it using a chart or diagram?

6. Mention the key variables and how they are measured.

7. Why did you perform the K-S test? Are they not continuous variables?

8. Why only the Baidu search was used? Why not Google search?

Reviewer 1 ·

Basic reporting

In this study, the researchers aimed to study the relationship between secretory otitis media (SOM) and its neighboring organ diseases, as well as the changes in their incidence during the COVID-19 epidemic in 2020 based on Internet big data.

Experimental design

The study is done in 2020 and it is quite old. For the investigators, is possible to refine the article and add the latest data? The authors can use the same keywords and do the search strategy and see the results. Maybe only some new articles can be added. These can add value to the article.
In the abstract, it is better to add the conclusion in more detail.

The researchers used Baidu and Google to search the article. There are other reliable scientific sources such as Scopus, and Web of science. So why they used only two, please give more clarification.

Validity of the findings

In the discussion, details about COVID-19 are added which are not necessary. They need to focus on the relationship between secretory otitis media (SOM) and its neighboring organ diseases. They can add some of their incidence in COVID-19.

·

Basic reporting

no comment except enriched the references and language pattern as more professional.

Experimental design

Found highly technical and maintained the ethical standard of the Journal.

Validity of the findings

Statistically sound & controlled.

Additional comments

Avoid the worlds like I, we as used in the Materials & Methods.
Make more professional/research language as such-- this paper, the study instead of using I or we.

Reviewer 3 ·

Basic reporting

Raw data contains some Chinese features.

Experimental design

The knowledge gap needs to be clarified in more detail.

Validity of the findings

Data analysis methods need to be clarified rigorously.

Additional comments

A tabular format should be used to present the results of the analysis. Based on research findings, a recommendation is needed.

·

Basic reporting

For the raw data, some information was not written in English. I suggest there should be an interpretation in English of the Chinese words or figures as the case may be.

Experimental design

No comment

Validity of the findings

No comment

Additional comments

I commend the authors for their detailed work which was written professionally and to the standards of the journal.

---

## Round 0.2 · accepted · Accept

Thanks for making the changes as suggested.